# Charge order driven by multiple-Q spin fluctuations in heavily electron-doped iron selenide superconductors

Ziyuan Chen [1,8], Dong Li [2,3,8], Zouyouwei Lu [2,3], Yue Liu [2,3], Jiakang Zhang [1], Yuanji Li [1], Ruotong Yin [1], Mingzhe Li [1], Tong Zhang [4,5,6], Xiaoli Dong [2,3,7], Ya-Jun Yan [1] ✉ & Dong-Lai Feng [1,5,6] ✉

Intertwined spin and charge orders have been widely studied in high-temperature superconductors, since their fluctuations may facilitate electron pairing; however, they are rarely identified in heavily electron-doped iron selenides. Here, using scanning tunneling microscopy, we show that when the superconductivity of $(Li_{0.84}Fe_{0.16}OH)Fe_{1-x}Se$ is suppressed by introducing Fe-site defects, a short-ranged checkerboard charge order emerges, propagating along the Fe-Fe directions with an approximately $2a_{Fe}$ period. It persists throughout the whole phase space tuned by Fe-site defect density, from a defect-pinned local pattern in optimally doped samples to an extended order in samples with lower $T_c$ or non-superconducting. Intriguingly, our simulations indicate that the charge order is likely driven by multiple-Q spin density waves originating from the spin fluctuations observed by inelastic neutron scattering. Our study proves the presence of a competing order in heavily electron-doped iron selenides, and demonstrates the potential of charge order as a tool to detect spin fluctuations.

Intertwined spin and charge orders have been widely observed in cuprate and iron-based superconductors and their parent compounds[1–24]. As superconductivity and these orders are often governed by the same interactions, they could be intimately related. For example, in the static form, spin and charge orders are sometimes considered as competing orders to superconductivity, while in their dynamic form, the spin/charge/orbital fluctuations of some of these orders are suggested to mediate the superconducting pairing[1–5,25], so the ordered phases are sometimes considered as the parent phases.

Antiferromagnetic and nematic order are the main form of intertwined orders in iron-based superconductors with both electron and hole Fermi surfaces, such as iron pnictides and bulk Fe(Se, Te)[13–24]. The nematic spin and orbital fluctuations are related to the scattering between electron and hole Fermi surfaces, which facilitate superconducting pairing in various theories[26–29]. However, such correspondence among intertwined orders, fluctuation and Fermi surface topology fails in heavily electron-doped FeSe superconductors, such as $K_{1-x}Fe_{2-y}Se_2$, $(Li_{0.84}Fe_{0.16}OH)Fe_{1-x}Se$, and monolayer FeSe/SrTiO$_3$, where only electron Fermi surfaces are present. Although a $2 \times 2$ charge order with respect to the Fe lattice was found in $K_xFe_{2-y}Se_2$[30,31], the ordering wave vector does not fit their electron-only Fermi surface topology, and its relation to superconductivity is uncertain. Moreover, a $\sqrt{5} \times \sqrt{5}$ Fe-vacancy order was observed in the insulating regions of $K_{1-x}Fe_{2-y}Se_2$, and its relation to superconductivity is remote[5]. On the other hand, inelastic neutron scattering (INS) experiments found that the low energy spin fluctuations are quite concentrated around

[1]School of Emerging Technology and Department of Physics, University of Science and Technology of China, Hefei 230026, China. [2]Beijing National Laboratory for Condensed Matter Physics, Institute of Physics, Chinese Academy of Sciences, Beijing 100190, China. [3]School of Physical Sciences, University of Chinese Academy of Sciences, Beijing 100049, China. [4]Department of Physics, State Key Laboratory of Surface Physics and Advanced Material Laboratory, Fudan University, Shanghai 200438, China. [5]Collaborative Innovation Center of Advanced Microstructures, Nanjing 210093, China. [6]Shanghai Research Center for Quantum Sciences, Shanghai 201315, China. [7]Songshan Lake Materials Laboratory, Dongguan, Guangdong 523808, China. [8]These authors contributed equally: Ziyuan Chen, Dong Li. ✉e-mail: yanyj87@ustc.edu.cn; dlfeng@ustc.edu.cn

($\pi \pm \delta\pi$, $\pi$) and ($\pi$, $\pi \pm \delta\pi$) in $Rb_xFe_{2-y}Se_2$ and $(Li_{0.84}Fe_{0.16}OD)Fe_{1-x}Se$ with $\delta \sim 0.5$ and 0.38 (refs. [32–35]), respectively, which seems to be unrelated to any known order in these systems. Spin fluctuation is widely accepted as the clue of electron pairing in cuprate and iron pnictide superconductors, but it is still under strong debate in heavily electron-doped FeSe compounds[36–42].

Recently, it was shown that the superconducting transition temperature ($T_c$) of $(Li_{0.84}Fe_{0.16}OH)Fe_{1-x}Se$ can be gradually suppressed by increasing the Fe-site defect concentration ($x$) in FeSe layers[43–48]. It thus provides a venue to search for the possible intertwined orders in heavily electron-doped iron selenides, especially the possible magnetic order in the parent compound. Here, we systematically studied $(Li_{0.84}Fe_{0.16}OH)Fe_{1-x}Se$ films with various $T_c$ values by using low-temperature scanning tunneling microscopy (STM). With an increased $x$, we find that the superconducting gap feature is gradually suppressed, and a static checkerboard charge pattern emerges firstly as a defect-pinned state and subsequently extends to the whole defect-free regions, illustrating a competing behavior with superconductivity. Combining with our simulations, the charge order is likely driven by multiple-$\mathbf{Q}$ spin density waves originating from the spin fluctuations observed by INS experiments. Our results demonstrate the potential of charge order as a tool to detect spin fluctuations, and suggest that spin fluctuations may also play an important role in heavily electron-doped FeSe-based superconductors.

## Results

### Influence of Fe-site defect on superconductivity

Figure 1a–d shows the typical topographic images of FeSe-terminated surfaces for three $(Li_{0.84}Fe_{0.16}OH)Fe_{1-x}Se$ films with $T_c$ of 42 K, 28 K, and 8 K, and another non-superconducting one, respectively (Supplementary Fig. 1). There is no obvious difference except for different amounts of dumbbell-shaped defects located at Fe-sites. These Fe-site defects can be roughly classified into two types according to their impurity potentials, Fe vacancies (type-I) or substitutional defects (type-II), and they both suppress superconductivity[40,49–52] (see Supplementary Fig. 2 for more details). The statistical Fe-site defect concentration, $x$, increases from 0.54% in Fig. 1a to 6.35% in Fig. 1d. Figure 1e shows the typical $dI/dV$ spectra within ±100 meV measured on defect-free regions of these $Fe_{1-x}Se$ surfaces. These spectra display similar line shape, with a relatively low tunneling conductance near $E_F$ but which increases rapidly above +70 meV and below −50 meV. As reported previously, these $dI/dV$ upturns are from the onsets of an unoccupied electron band and an occupied hole band at the Brillouin zone (BZ) center, respectively; and the states near $E_F$ are contributed by electron bands at $M$ points of BZ[53,54]. The almost identical $dI/dV$ spectra observed for different defect concentrations suggest that Fe-site defects have little influence on carrier doping.

The $dI/dV$ spectra within ±40 meV measured in defect-free FeSe regions are shown in Fig. 1f (spatially averaged spectra, and see Supplementary Fig. 3 for point spectra). For $x = 0.54$%, a full superconducting gap with two pairs of coherence peaks at ±9 meV and ±15 meV is observed. With increasing $x$, the coherence peaks are weakened quickly, and the gap feature becomes broadened. Such overall suppression of superconductivity by Fe-site defects is similar to that observed in superconducting $KFe_2Se_2$ films[55], confirming that Fe-site defect concentration is the key parameter controlling the superconductivity of $(Li_{0.84}Fe_{0.16}OH)Fe_{1-x}Se$[45]. Meanwhile, the spectral weight within the whole energy range of ±40 meV is also strongly suppressed with increased $x$, and a broad V-shaped spectrum with almost zero density of states

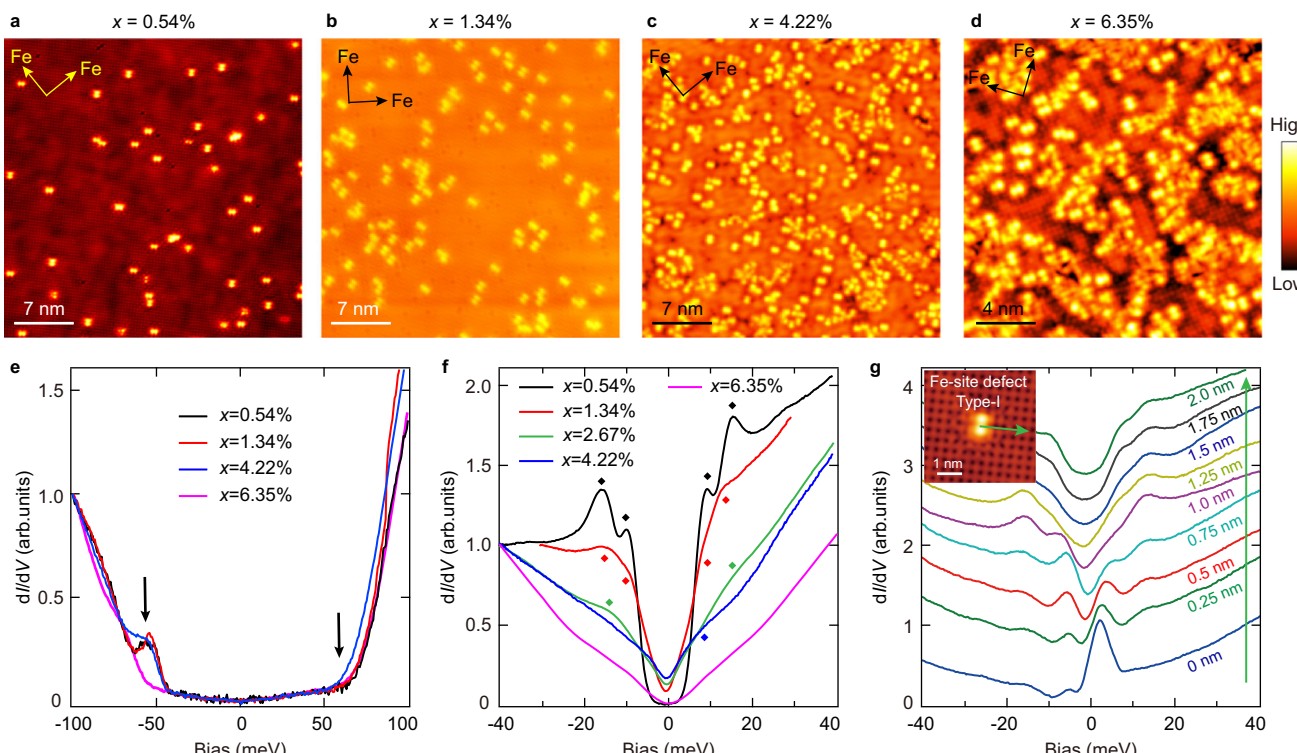

**Fig. 1 | Typical topographic images and $dI/dV$ spectra for $Fe_{1-x}Se$ regions with varying $x$ values. a–d** Atomically resolved topographic images of four $Fe_{1-x}Se$ regions. Measurement conditions: **a** $V_b = 0.5$ V, $I_t = 60$ pA; **b** $V_b = -0.55$ V, $I_t = 20$ pA; **c** $V_b = 0.1$ V, $I_t = 10$ pA; **d** $V_b = 0.1$ V, $I_t = 50$ pA. **e** Typical $dI/dV$ spectra within ±100 meV collected in defect-free FeSe regions ($V_b = 100$ mV, $I_t = 80$ pA, $\Delta V = 2$ mV). Black arrows indicate the onsets of energy bands. **f** Averaged $dI/dV$ spectra within ±40 meV collected in defect-free FeSe regions. The diamonds mark the positions of coherent peaks. Measurement conditions: $V_b = 30$ mV, $I_t = 120$ pA, $\Delta V = 0.1$ mV for $x = 1.34$%; $V_b = 40$ mV, $I_t = 120$ pA, $\Delta V = 0.1$ mV for other $x$ values. **g** Evolution of $dI/dV$ spectra across a type-I Fe-site defect in the $Fe_{1-x}Se$ region with $x = 0.54$%, taken along the green arrow in the inset. The distance from the spatial locations where the spectra are collected to the defect center is indicated and color-coded with the spectra. Inset: topographic image of a type-I Fe-site defect.

(DOS) around $E_F$ shows up for the non-superconducting $x = 6.35\%$ sample, which evidence a spectral gap opening. Such a partial gap is likely related to the formation of a charge order discussed in the next section. As shown in Fig. 1g, the superconducting gap is almost completely suppressed and strong in-gap states are observed on the type-I Fe-site defects, while the superconducting gap recovers beyond 1.5 nm away from the defect center.

### Observation of the checkerboard charge patterns

d$I$/d$V$ maps were acquired on different $Fe_{1-x}Se$ regions to reveal the local density of state (LDOS) distribution. Figure 2a, b shows the typical d$I$/d$V$ maps collected in the $Fe_{1-x}Se$ ($x = 0.54\%$) region in Fig. 1a (see Supplementary Fig. 4 for additional datasets under other energies). One can see clear quasiparticle interference (QPI) patterns around type-I Fe-site defects, which contribute to nine ring-like dispersive

features around the BZ center and Bragg spots ($\mathbf{q}_{Fe}$ and $\mathbf{q}_{Se}$) in the fast Fourier transforms (FFT) image (Fig. 2c). These QPI patterns arise from intra- and interpocket scatterings of the electron pockets at $M$ points of BZ, which have been widely observed in heavily electron-doped FeSe materials[40–42,56,57]. Intriguingly, besides the common QPI patterns, we can resolve additional FFT features around $\frac{1}{2}\mathbf{q}_{Fe}$ (labeled as $\mathbf{q}_{2Fe}$) as indicated by the yellow arrows in Fig. 2c. This feature is nondispersive over the measured energy range of ±45 meV, as indicated by the red dashed line in Fig. 2d. It is not clear near the Fermi energy though, which is likely due to the strong in-gap state induced by the impurity.

The LDOS distribution around type-II Fe-site defects exhibits a subtle structure, as enclosed in the yellow dashed circles in Fig. 2a, b. As more clearly illustrated in Fig. 2e, f, it is a checkerboard pattern oriented similarly along the two Fe-Fe lattice directions with a period approximately twice that of the Fe lattice ($2a_{Fe}$), which naturally leads

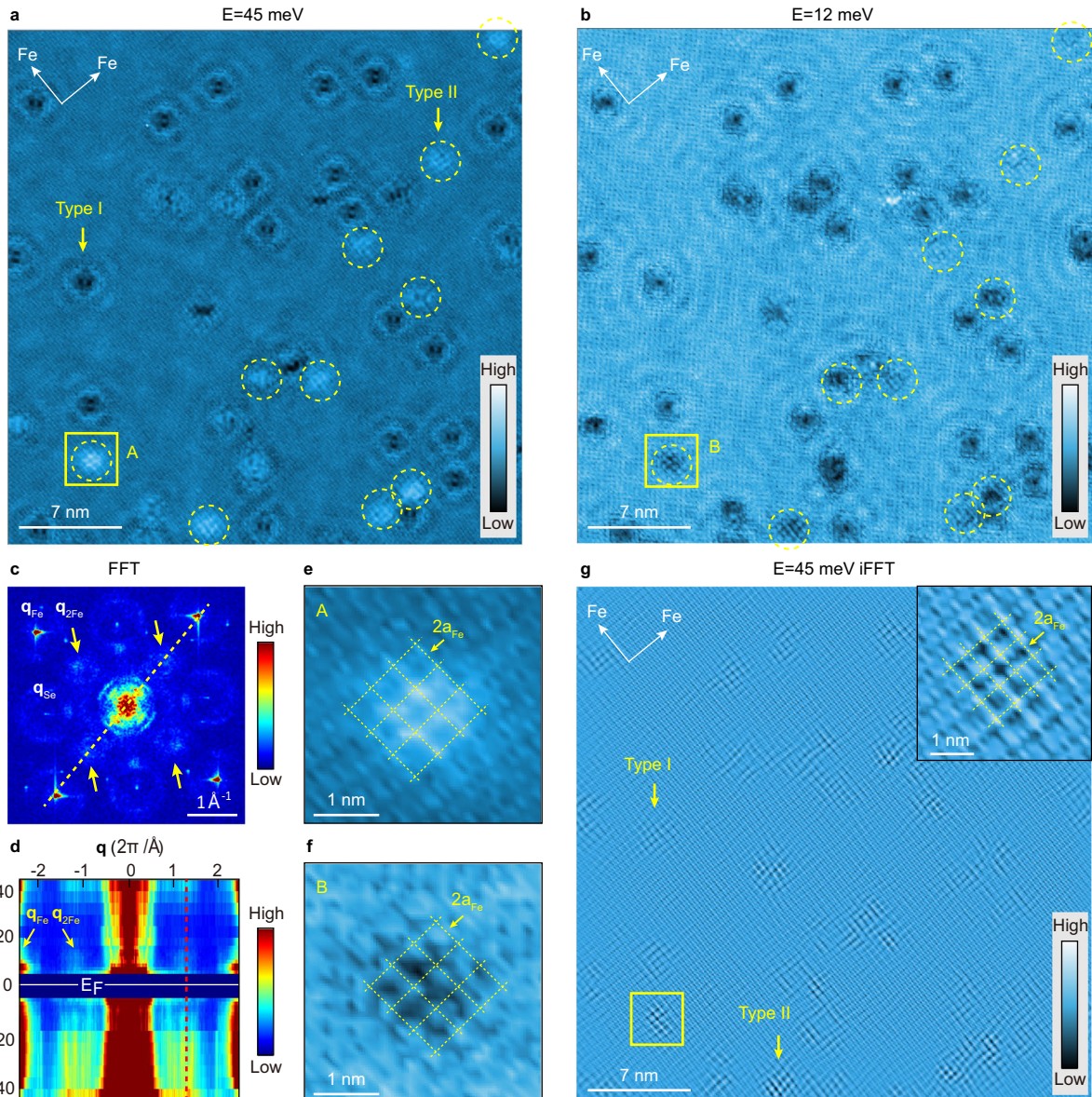

**Fig. 2 | Spatial LDOS modulations in the $Fe_{1-x}Se$ region with $x = 0.54\%$.**
**a**, **b** Typical d$I$/d$V$ maps ($V_b = 45$ mV, $I_t = 50$ pA, $\Delta V = 2$ mV). Type-II Fe-site defects are enclosed by the yellow dashed circles, while the others are the type-I defects. **c** FFT image of the d$I$/d$V$ map shown in (**a**). Additional FFT features around $\mathbf{q}_{2Fe}$ are marked by yellow arrows. **d** FFT line cuts extracted along the yellow dashed line in (**c**), taken at various energies and shown in false color. The red dashed line indicates

the nondispersive nature of $\mathbf{q}_{2Fe}$ pattern. **e**, **f** Enlarged images of areas A and B marked by yellow boxes in (**a**) and (**b**), with grids of a $2a_{Fe}$ period superimposed. **g** iFFT image with features around $\mathbf{q}_{2Fe}$ and their second-order components around $\mathbf{q}_{Fe}$ considered. Insert: enlarged image of the area marked by yellow box in (**g**), with a grid of a $2a_{Fe}$ period superimposed.

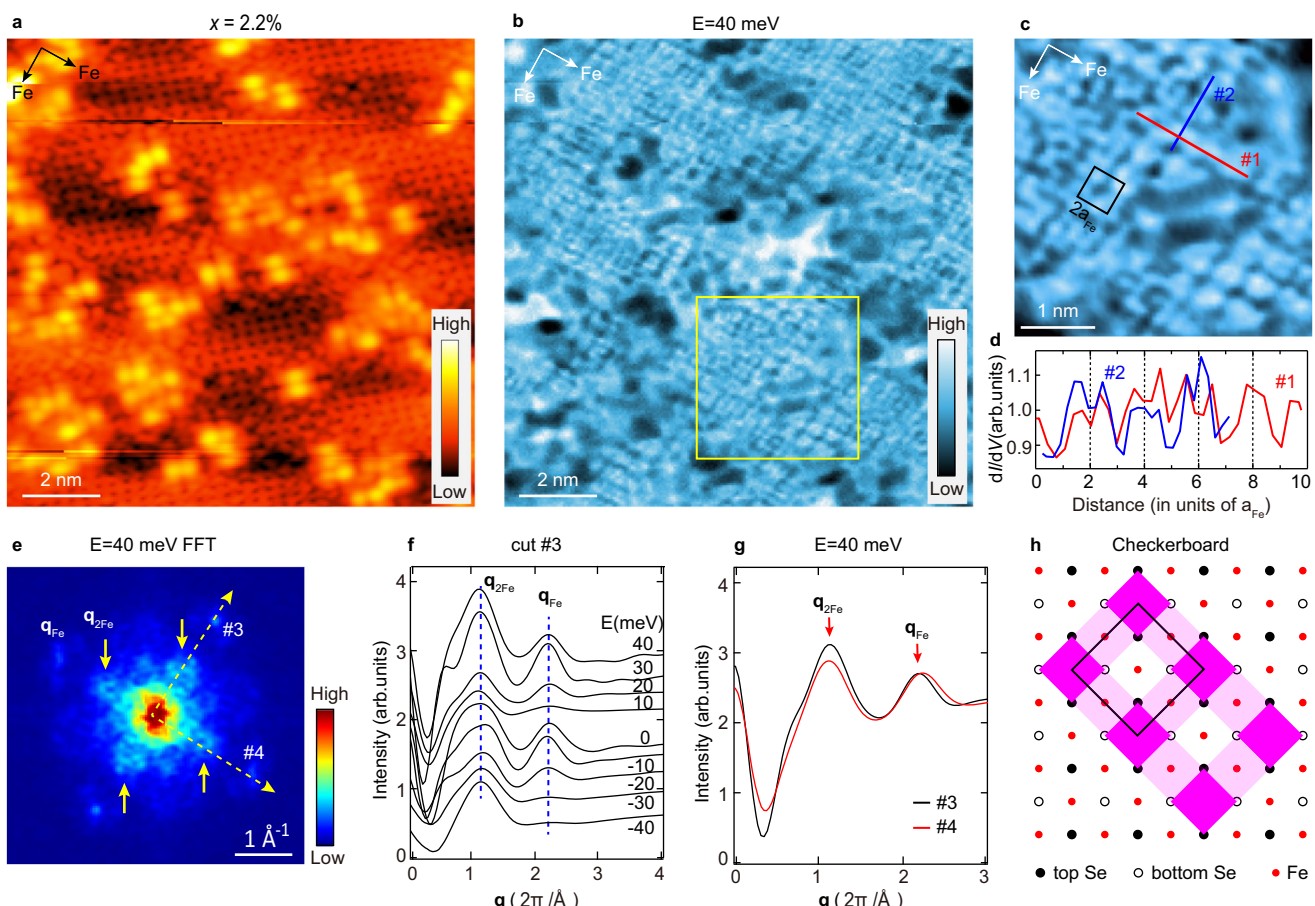

**Fig. 3 | Spatial LDOS modulations in an $Fe_{1-x}Se$ region with $x = 2.2\%$. a** Atomically resolved topographic image ($V_b = 40$ mV, $I_t = 20$ pA). **b** Typical d$I$/d$V$ map at $E = 40$ meV (setpoint: $V_b = 40$ mV, $I_t = 50$ pA, $\Delta V = 2$ mV). **c** Enlarged image of the area marked by yellow box in (**b**). The black box shows the unit cell of the checkerboard pattern with a $2a_{Fe}$ period. **d** Spatial LDOS profiles taken along cuts #1 and #2 in (**c**). **e** FFT image of the d$I$/d$V$ map in (**b**). Additional FFT features around $q_{2Fe}$ are marked by yellow arrows. **f** FFT line cuts extracted along cut #3 in (**e**) under various energies. A fitted Lorentz-shaped background was subtracted for each curve, and the curves were shifted vertically for clarity. The two blue dashed lines indicate the positions of $q_{Fe}$ and $q_{2Fe}$. **g** Comparison of FFT line cuts at $E = 40$ meV extracted along cuts #3 and #4 in (**e**). **h** Schematic illustration of the fine structure of the checkerboard pattern atop the Fe and Se lattices. Unit cell of this structure is indicated by black box. The magenta squares represent higher LDOS observed by STM, and the dark and light colors show difference in LDOS intensity.

to FFT intensities around $q_{2Fe}$. We conducted inverse FFT (iFFT) of the features around $q_{2Fe}$ and their second-order components around $q_{Fe}$ to remove the interference of QPI signals, which gives Fig. 2g and Supplementary Fig. 4. The checkerboard patterns now became more distinguishable and show up around both type-I and type-II Fe-site defects. As moving away from a defect, the checkerboard pattern weakens quickly and disappears beyond ~1.5–2.0 nm, which is comparable to the length scale of superconducting gap suppression around Fe-site defects (Fig. 1g). One may wonder if the pinned checkerboard pattern is due to the Yu-Shiba-Rusinov (YSR) states induced by Fe-site defects. We have excluded this possibility by carefully examining the spatial distribution of the YSR states, which turns out to behave very differently (See Supplementary Fig. 5). Moreover, as shown later for the higher $x$ cases (Figs. 3 and 4), the checkerboard pattern still exists and extends into defect-free regions (even in the non-superconducting samples), implying that the checkerboard pattern is a static order instead of YSR states. Consistently, such a checkerboard pattern is absent near Se vacancies, where superconductivity is intact (Supplementary Fig. 6). As STM measures tunneling-matrix-element weighted LDOS, the checkerboard pattern could reflect either a charge density modulation or an orbital reconstruction.

Figure 3 examines the spatial LDOS modulation of an $Fe_{1-x}Se$ region with $x = 2.2\%$. The topography of this region is shown in Fig. 3a,

and the corresponding d$I$/d$V$ maps are displayed in Fig. 3b and Supplementary Fig. 7. A checkerboard pattern is ubiquitously present in the defect-free areas over the measured energy range of ±40 meV; it propagates along the two Fe-Fe lattice directions and exhibits a $2a_{Fe}$ period, as illustrated by the LDOS profiles in Fig. 3c, d. Accordingly, the FFT intensities at $q_{2Fe}$ are greatly enhanced (Fig. 3e), corresponding to the growing checkerboard patterns with increasing $x$. The checkerboard pattern is nondispersive as indicated by the local maxima at $q_{2Fe}$ of the FFT line cuts under varying energies in Fig. 3f; it is $C_4$-symmetric as can be directly seen from the FFT images and is further confirmed by the similar spectral weight of FFT features around $q_{2Fe}$ along the two Fe-Fe lattice directions at $E = 40$ meV (Fig. 3g). Moreover, the checkerboard pattern is more clearly resolved in the iFFT images based on the features around $q_{2Fe}$ and their second-order components around $q_{Fe}$ (see Supplementary Fig. 7 and Supplementary Fig. 8). Such an extended checkerboard pattern was also observed in another $Fe_{1-x}Se$ region with a similar $x \sim 1.8\%$ (Supplementary Fig. 9). By comparing the atomically resolved Se lattice in Fig. 3a and the LDOS distribution of the checkerboard pattern in Fig. 3c, we infer a possible fine structure of the checkerboard order atop the Fe and Se lattice as sketched in Fig. 3h —neighboring FeSe$_4$ tetrahedrons differ in LDOS intensity, exhibiting a period of $2a_{Fe}$ along the two Fe-Fe directions.

Figure 4 shows the results of $Fe_{1-x}Se$ regions with higher $x$ values. With further increased defect density (Fig. 4a, e), the checkerboard

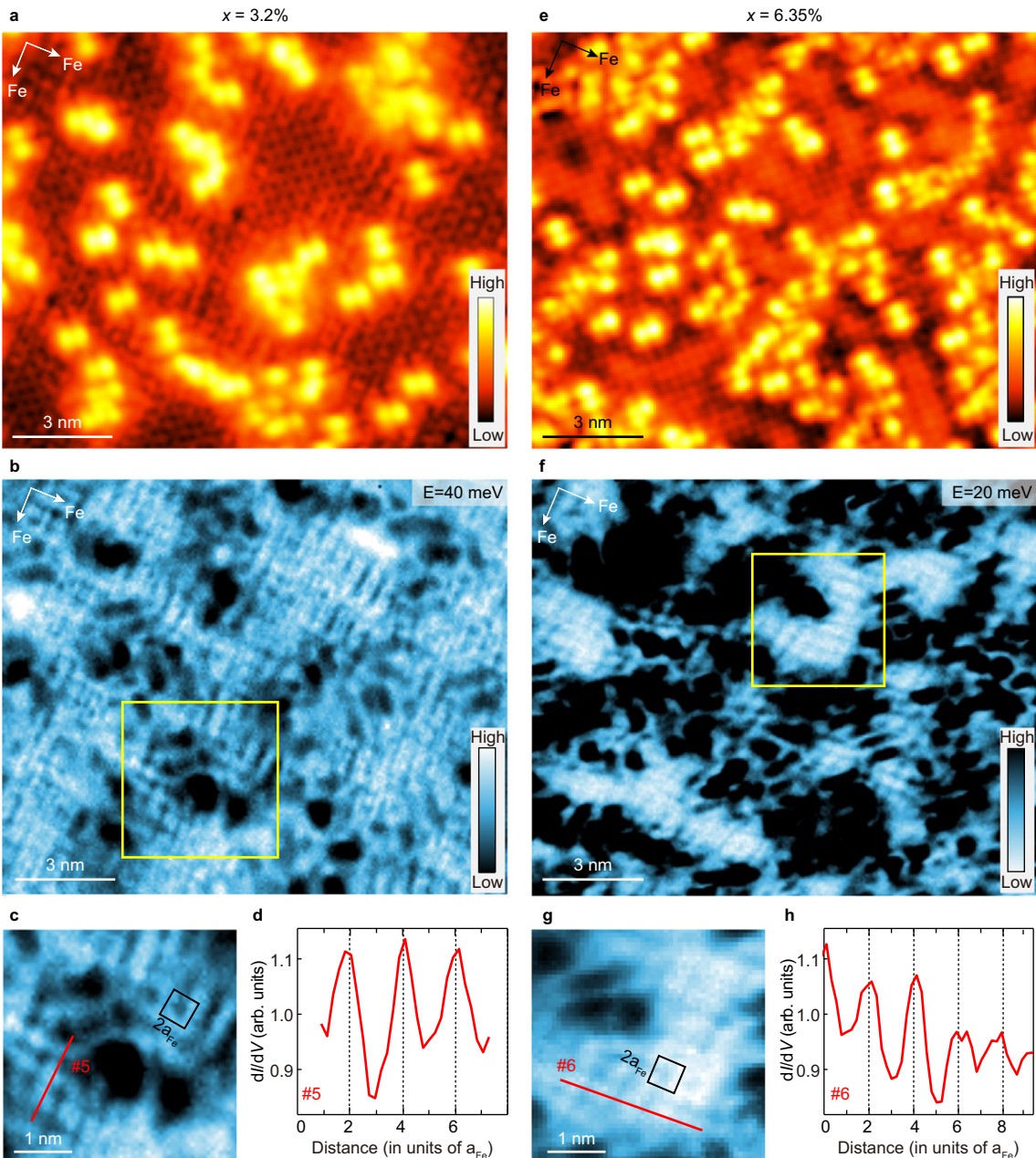

**Fig. 4 | Spatial LDOS modulations in $Fe_{1-x}Se$ regions with $x = 3.2\%$ and $6.35\%$.**
**a**, **b** Atomically resolved topographic image and corresponding $dI/dV$ map at $E = 40$ meV for an $Fe_{1-x}Se$ region with $x = 3.2\%$. Measurement conditions for **a**: $V_b = 40$ mV, $I_t = 50$ pA and **b**: $V_b = 40$ mV, $I_t = 50$ pA, $\Delta V = 2$ mV. **c** The enlarged image of the area marked by the yellow box in (**b**). The black box shows the unit cell of the charge order with a $2a_{Fe}$ period. **d** Spatial LDOS profiles taken along cut #5 in (**c**).

**e**, **f** Atomically resolved topographic image and corresponding $dI/dV$ map at $E = 20$ meV for an $Fe_{1-x}Se$ region with $x = 6.35\%$. Measurement conditions for **e**: $V_b = 100$ mV, $I_t = 50$ pA and **f**: $V_b = 100$ mV, $I_t = 50$ pA, $\Delta V = 10$ mV. **g** The enlarged image of the area marked by the yellow box in (**f**). The black box shows the unit cell of the charge order with a $2a_{Fe}$ period. **h** Spatial LDOS profiles taken along cut #6 in (**g**).

charge pattern persists in the defect-free FeSe regions and dominates even in the non-superconducting samples with $x = 6.35\%$, as shown in Fig. 4b, f and Supplementary Figs. 10–13. Figure 4c, g shows the enlarged images of the areas indicated by yellow boxes in Fig. 4b, f, exhibiting clear charge modulations with $2a_{Fe}$-period for both cases, which are further confirmed by the LDOS profiles shown in Fig. 4d, h. Overall, the static checkerboard charge pattern is a general feature of heavily electron-doped FeSe, regardless of the defect density and superconducting transition temperature.

Sometimes we can see unidirectional charge stripes in the vicinity of Fe-site defects, as shown in Supplementary Fig. 14. It seems likely that in a small sample region, a $2a_{Fe}$-period modulation is preserved

along one of the Fe-Fe lattice directions, whereas it is smeared in the perpendicular direction. The appearance of unidirectional charge stripes seems strongly depend on the local environment. Its origin is not clear now, and we speculate that local strains induced by gathered Fe-site defects may be responsible.

Here we summarize how the Fe-site defect concentration dominates the behavior of superconductivity and charge order in $(Li_{0.84}Fe_{0.16}OH)Fe_{1-x}Se$. At first, when $x$ is low, $T_c$ can be as high as 42 K. Superconductivity is only suppressed at the defect sites[40,50–52], and a checkerboard pattern is pinned there (Fig. 2). With increased $x$, the overall superconductivity is suppressed and $T_c$ drops, the checkerboard pattern extends to the whole defect-free areas (Fig. 3 and

Fig. 4b) and persists even after the superconductivity disappears when $x > 5\%$ (Fig. 4f). Such an evolution clearly shows that the checkerboard order competes with superconductivity, which flourishes when superconductivity is partly or fully suppressed by Fe-site defects. The formation of such charge orders could explain the suppression of spectral weight around $E_F$ for higher $x$ (Fig. 1f). Moreover, we find the checkerboard charge order is short-ranged and the estimated in-plane correlation length is approximately $4a_{Fe}$ ~$5a_{Fe}$, as discussed in detail in Supplementary Fig. 15. Besides, the period of the checkerboard charge order is doping independent and maintains a value of ~$2a_{Fe}$ with varying $x$, which is consistent with the unaffected carrier doping level and the doping independent band structures.

## Simulations of spin and charge orders

The ordering wave vector of the checkerboard order does not match the nesting vectors between Fermi surface pockets in $(Li_{0.84}Fe_{0.16}OH)$ $Fe_{1-x}Se$. Therefore, one needs to examine how Fe-site defects may induce the observed charge orders. Although Fe-site defects in $Fe_{1-x}Se$ layer do not alter the itinerant carriers effectively, they have been shown to expand the in-plane lattice and reduce the out-of-plane lattice parameter[45,47], so that the electronic structure will be altered. Moreover, the Fe oxidation state changes from slightly below +2 to higher values[45,58]; thus, the Fe orbital configuration is driven toward the half-filled $3d^5$ state. Consequently, both the electron correlation and the local magnetic moment of Fe ions in the $Fe_{1-x}Se$ layer would be enhanced[59,60]. We notice that weak magnetic behavior was observed in superconducting $(Li_{0.84}Fe_{0.16}OH)Fe_{1-x}Se$[43], and an enhanced local Fe moment was deduced from a Curie-Weiss-like magnetic susceptibility in insulating $(Li_{0.84}Fe_{0.16}OH)Fe_{0.7}Se$[58].

The enhanced electron correlation and local moment would promote magnetism; thus, the magnetic fluctuations may be stabilized into static order. A $2 \times 2$ charge order in the FeSe layer was reported in $K_{1-x}Fe_{2-y}Se_2$ before[30,31], which was considered to be related to a block-antiferromagnetic order suggested in the parent compound of $K_{1-x}Fe_{2-y}Se_2$ (ref. 61), corresponding to a Bragg spot located at $(\pi/2, \pi/2)$. For $(Li_{0.84}Fe_{0.16}OH)Fe_{1-x}Se$, there are no low-energy spin fluctuations at $(\pi/2, 0)$, $(0, \pi/2)$, and $(\pi/2, \pi/2)$, so the observed charge orders are unlikely to be driven by stabilizing spin fluctuations at these momenta. Instead, INS measurements on optimally doped $(Li_{0.84}Fe_{0.16}OD)Fe_{1-x}Se$ have found low energy magnetic excitations centered around four momenta $(\pi \pm \delta\pi, \pi)$ and $(\pi, \pi \pm \delta\pi)$ $(\delta \sim 0.38)$ (ref. 32) with similar spectral weights, as reproduced in Fig. 5a. We note that such a momentum distribution is broad, covering commensurate values such as $\delta = 0.5$, which corresponds to the scatterings between certain sectors of the Fermi surfaces with distinct orbital characters (Fig. 5b). The horizontal $Q_3, Q_4, Q'_3, Q'_4$ and vertical $Q_1, Q_2, Q'_1, Q'_2$ patterns correspond to the scatterings between the Fermi surface sectors of $d_{xy}$ orbitals with that of $d_{xz}$ or $d_{yz}$ orbitals, respectively. In heavily electron-doped FeSe, the lattice is $C_4$-symmetric and the $d_{xz}$ or $d_{yz}$ orbitals are degenerate, thus the scatterings between $d_{xy}$ orbitals and $d_{xz}$ or $d_{yz}$ orbitals are degenerate as well, leading to the same spectral weights of different magnetic excitations. Therefore, we will consider the superposition of multiple magnetic excitations with equal weightage.

Assuming that the spin fluctuation distributions are similar in our samples and that the $\delta = 0.5$ spin fluctuations could be stabilized into commensurate static orders by Fe-site defects, we could qualitatively simulate the resulting spin density as $I_s = \cos(Q_i \cdot r)$ (note that $Q_i$ and $-Q_i$ are equivalent). Moreover, it has been shown that for a collinear ferromagnet or antiferromagnet, the charge redistribution induced by spin-orbit coupling depends on the spin axis instead of its direction[62]; as a result, the period of the charge modulation is half of that of the spin modulation. Such a relation between spin and charge orders has been observed in cuprates[7–11], MnP[63], Cr crystals[64], and also predicted for iron-based superconductors[65]. Therefore, the charge

order driven by such a spin density wave (SDW) can be qualitatively simulated using $I_c = |I_s| = |\cos(Q_i \cdot r)|$.

First, a simple SDW and the corresponding charge order are simulated by choosing one of the $Q_i$ vectors ($Q_1 = (0.5, 0.75)$, $Q_2 = (0.5, 0.25)$, $Q_3 = (0.75, 0.5)$, $Q_4 = (0.25, 0.5)$, $Q'_1 = (-0.5, 0.75)$, $Q'_2 = (-0.5, 0.25)$, $Q'_3 = (-0.75, 0.5)$, $Q'_4 = (-0.25, 0.5)$ in the reciprocal lattice unit (r.l.u.)) shown in Fig. 5a. The results are plotted in Supplementary Fig. 16a3-d3, and the tilted charge stripes differ completely from the experimental patterns. Alternatively, a spin order could result from a coherent addition of several SDWs with different wave vectors, and a double-$Q$ SDW was proposed to account for the spin order in $Sr_{0.63}Na_{0.37}Fe_2As_2$ (ref. 66). A multiple-$Q$ SDW could be simulated by a coherent addition of various SDWs: $I_s = \sum \cos(Q_i \cdot r)$. Supplementary Fig. 16e2-h2 and e3-h3 present double-$Q$ SDWs and corresponding charge orders simulated with various pairs of $Q_i$ vectors. Most of these patterns fail to reproduce the experiments in terms of the period and orientation, except the one based on $Q_4$ and $Q'_4$ that reproduces the period and orientation of vertical stripes (see Supplementary Fig. 16h3). Furthermore, considering the complete magnetic excitation patterns in Fig. 5a, it is more natural to include the four horizontal or vertical $Q_i$ vectors. The resulting quadruple-$Q$ SDW based on $Q_1, Q_2, Q'_1, Q'_2$ is plotted in Fig. 5c and has a $4a_{Fe} \times 2a_{Fe}$ unit cell. The corresponding charge order in Fig. 5d shows the horizontal stripes with a $2a_{Fe} \times a_{Fe}$ unit cell. Likewise, the vertical charge stripes in Fig. 5e correspond to the coherent addition of SDWs with $Q_3, Q_4, Q'_3, Q'_4$. The charge distribution driven by the total eight momenta is shown in Fig. 5f, which is rotated 45° with a $2\sqrt{2}a_{Fe} \times 2\sqrt{2}a_{Fe}$ unit cell, failing to reproduce the experiment. On the other hand, a direct overlap of the vertical and horizontal stripes could qualitatively reproduce the checkerboard pattern (Fig. 5g). This may also explain the observed local unidirectional charge stripes in the vicinity of Fe-site defects, when one set of stripes is stronger than the other (Supplementary Fig. 16m3). Such a broken symmetry may be caused by various reasons. For example, the local Fe-site defect distribution may favor scattering between the $d_{xz}$ and $d_{xy}$ orbitals rather between the $d_{yz}$ and $d_{xy}$ orbitals (Fig. 5b). In such a multiple-$Q$ density wave scenario, the $2a_{Fe} \times 2a_{Fe}$ checkerboard order and $2a_{Fe} \times 1a_{Fe}$ charge stripe found in $(Li_{0.84}Fe_{0.16}OH)Fe_{1-x}Se$ could be naturally explained. Figure 5h illustrates that the simple simulations of the double-$Q$ ($Q_4$ and $Q'_4$) and quadruple-$Q$ ($Q_3, Q_4, Q'_3,$ and $Q'_4$) charge patterns could qualitatively reproduce the periodic patterns of the charge orders. Because of the complexity in tunneling matrix element, it is still premature to decide which is correct. This calls for further theoretical investigations to understand the fine structures of the checkerboard order and localized charge stripes, and to distinguish between double-$Q$ and quadruple-$Q$ SDW scenarios.

## Discussion

Note that spin fluctuations are usually much stronger than charge fluctuations in low-dimensional correlated systems; thus, charge/orbital orders could form at higher temperatures than those of spin ordering[13,14,63]. Therefore, even though we observed charge order here, the spin order might still be dynamic. Long-range magnetic order has not been found in $(Li_{0.84}Fe_{0.16}OH)Fe_{1-x}Se$ by macroscopic measurements[32,33], here we indicate that when the superconductivity is effectively suppressed, the magnetic order or the enhanced magnetic fluctuations stabilized by local defects can be indirectly detected by the presence of charge orders. Future spin-resolved STM studies could further clarify the existence of local magnetic order around the Fe-site defects. If it exists, it should be viewed as the major competing order for superconductivity.

We may also examine other possible origins for the charge order, for example, if it is driven by phonon-softening effects. Ideally, inelastic x-ray scattering may be conducted to clarify this, but it is very

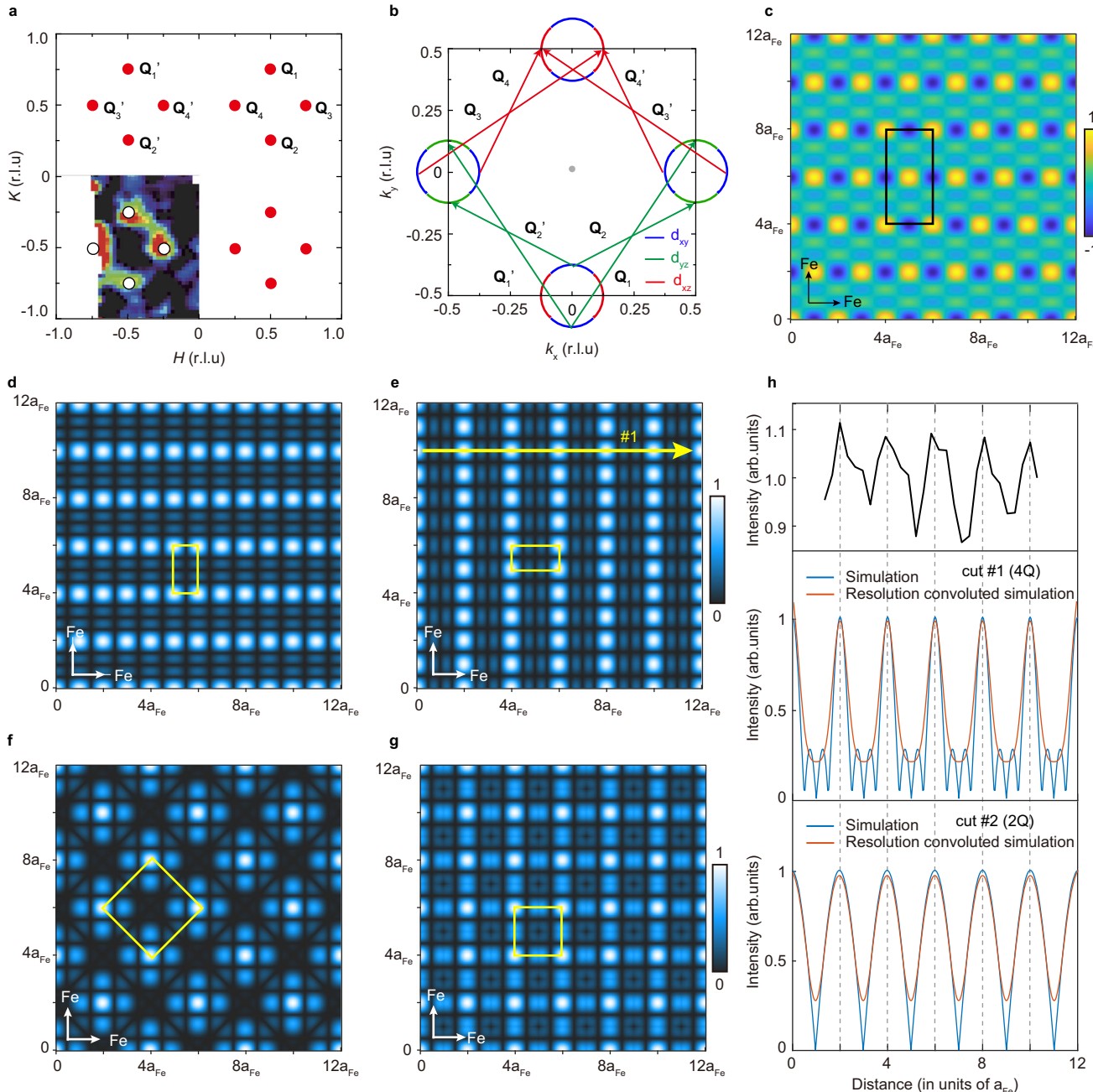

**Fig. 5 | Simulation of spin and charge patterns in real space. a** Momentum distribution of spin fluctuations in $(Li_{0.84}Fe_{0.16}OD)Fe_{1-x}Se$ measured by INS in momentum space. The red dots mark the commensurate spin fluctuation momenta, as shown in (**b**). **b** Cartoon of the corresponding scatterings between Fermi surface sections in the unfolded BZ for the momenta shown in (**a**). **c** The simulated SDW pattern based on four wave vectors: $Q_1 = (0.5, 0.75)$, $Q_2 = (0.5, 0.25)$, $Q_1' = (-0.5, 0.75)$, $Q_2' = (-0.5, 0.25)$ (in reciprocal lattice unit (r. l. u.)), with its unit cell indicated by the black box. **d–f** The simulated charge order patterns by considering **d**: $Q_1, Q_2, Q_1', Q_2'$, **e**: $Q_3 = (0.75, 0.5)$,

$Q_4 = (0.25, 0.5)$, $Q_3' = (-0.75, 0.5)$ and $Q_4' = (-0.25, 0.5)$, and **f**: eight wave vectors of $Q_i$ and $Q_i'$ ($i = 1 - 4$). **g** The charge order pattern obtained by an overlay of the patterns shown in (**d**) and (**e**). The unit cells of these patterns are indicated by the yellow boxes. **h** Comparison between typical LDOS profile of the charge stripe (upper panel) and the charge distributions from simulations based on quadruple-$Q$ ($Q_3, Q_4, Q_3', Q_4'$) SDW (middle panel, taken along cut #1 in **e**) and double-$Q$ ($Q_4$ and $Q_4'$) SDW (lower panel, taken along cut #2 in Supplementary Fig. 16h3), respectively. The orange and blue curves show the simulated charge profiles with and without convolution with spatial resolution.

challenging for thin film samples. However, we note that usually one could observe the charge order in topographic image in the case of charge density wave caused by electron-phonon interactions, while we did not observe it in our case. Disorder-induced local strain is another possible cause. However, it is unlikely, because the checkerboard pattern extends to the large defect-free regions for $x = 1.8\%$ and 2.2% (Fig. 3 and Supplementary Fig. 9), where local strain should be weak. Moreover, with increased $x$ values, disorder-induced local strain will

increase in principle, but the observed checkerboard pattern does not change.

To conclude, our results establish the comprehensive relations among the observed competing order, spin fluctuations, and Fermi surface topology in heavily electron-doped FeSe superconductors. The stabilized double- or quadruple-$Q$ spin fluctuations that scatter between different parts of the Fermi surface likely drive the observed checkerboard charge order, when the superconductivity is suppressed

by Fe-site defects. Our results favor spin fluctuations rather than nematic orbital fluctuations as the driving force of the charge order, although the orbital fluctuations were suggested to be as important as spin fluctuations in iron-based superconductors with both electron and hole Fermi surfaces, such as iron pnictides and bulk Fe(Se,Te). Therefore, spin fluctuations may play an important role in Cooper pair formation in $(Li_{0.84}Fe_{0.16}OH)FeSe$ and other heavily electron-doped FeSe-based superconductors. Moreover, our study suggests that the charge order revealed by STM can be used as a tool to detect the spin fluctuations or magnetic orders of the parent phase, which can be extended to other systems in the future.

## Methods

### Synthesis of $(Li_{0.84}Fe_{0.16}OH)Fe_{1-x}Se$ films and characterizations

High-quality single-crystalline $(Li_{0.84}Fe_{0.16}OH)Fe_{1-x}Se$ films were grown on $LaAlO_3$ substrates by a matrix-assisted hydrothermal epitaxy method[46,47]. Large crystals of $K_{0.8}Fe_{1.6}Se_2$ (nominal 245 phase) are grown in advance and used as the matrix for facilitating the hydrothermal epitaxial growth. By adjusting the synthesis temperature in the range from 120 to 180 °C, these samples with different $T_c$ from 42 K down to 0 K are available. The $(Li_{0.84}Fe_{0.16}OH)Fe_{1-x}Se$ films have thicknesses of 600–700 nm, determined on a Hitachi SU5000 scanning electron microscope. X-ray diffraction patterns were collected at room temperature on a 9 kW Rigaku SmartLab diffractometer using Cu $K\alpha_1$ radiation ($\lambda = 1.5405$ Å), with a $2\theta$ range of 5°–80° and $2\theta$ scanning steps of 0.01°. All magnetic susceptibility measurements were conducted on a Quantum Design MPMS-XL1 system.

### STM measurements

$(Li_{0.84}Fe_{0.16}OH)Fe_{1-x}Se$ films were mechanically cleaved at 78 K in ultrahigh vacuum with a base pressure better than $1 \times 10^{-10}$ mbar and immediately transferred into a UNISOKU cryogenic STM at $T = 4.2$ K. Pt-Ir tips were used after being treated on a clean Au (111) substrate. The d$I$/d$V$ spectra were collected by a standard lock-in technique with a modulation frequency of 973 Hz and a typical modulation amplitude $\Delta V$ of 1-2 mV at 4.2 K.

## Data availability

All the data supporting the findings of this study are provided within the article and its Supplementary Information files. All the raw data generated in this study are available from the corresponding author upon reasonable request.

## Code availability

All the data analysis codes related to this study are available from the corresponding author upon reasonable request.

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

## Acknowledgements

We thank Prof. Bingying Pan, Prof. Zhongxian Zhao, and Ms. Yining Hu for helpful discussions. This work is supported by the National Natural Science Foundation of China (Grants No. 12074363 (Y.J.Y.), No. 11790312 (D.L.F.), No. 11888101 (D.L.F.), No. 11774060 (Y.J.Y.), No. 12061131005 (X.L.D.), and No. 92065202 (T.Z.)), the National Key R&D Program of the MOST of China (Grants No. 2017YFA0303004 (T.Z.) and No. 2022YFA1403900 (X.L.D.)), the Innovation Program for Quantum Science and Technology (Grant No. 2021ZD0302800 (D.L.F.)), and the Strategic Priority Research Program of Chinese Academy of Sciences (Grants No. XDB33010200 (X.L.D.) and XDB25000000 (X.L.D.)).

## Author contributions

$(Li_{0.84}Fe_{0.16}OH)Fe_{1-x}Se$ films were grown by D.L., Z.L., and Y. Liu under the guidance of X.D.; STM measurements were performed by Z.C. and Y.Y.; The data analysis was performed by Z.C., Y.Y., T.Z., D.F., J.Z., Y. Li, R.Y., and M.L.; Y.Y. and D.F. coordinated the whole work and wrote the manuscript. All authors have discussed the results and the interpretation.

## Competing interests

The authors declare no competing interests.
