## [Peer Review File · Nature Communications]

Charge order driven by multiple-Q spin fluctuations in heavily electron-doped iron selenide superconductorsREVIEWER COMMENTS

Reviewer #1 (Remarks to the Author):

Chen Ziyuan and co-workers report on scanning tunneling microscopy and spectroscopy measurements on the surface of iron (Fe) doped LiFeOHFeSe. Using spectroscopic imaging at different energies they investigate the appearance of localized local density of states (LDOS) patterns in the vicinity of Fe defect sites on the sample surface. Upon increasing Fe defect concentration superconductivity becomes suppressed and the LDOS pattern become more dominant. Comparing their experimental observations with model calculations, the authors argue that Fe defects suppress superconductivity and promote the emergence of competing charge orders whose type depends on the doping level.

Competing orders in cuprates and iron-based superconductors is an active research area and the manuscript under review, therefore, touches on an interesting topic. For reasons that I will describe in the following, the conclusions drawn from the authors appear premature and are not yet sufficiently supported by experimental data in a way that would warrant publication of this work in Nature Communications. The following comments must be addressed before I can recommend publication in Nature Communications.

Comments:

1. If the increased Fe impurity density were to induce a charge ordered state (nematic or not), a spectral gap should develop in the dI/dV spectrum around Fermi energy. The dI/dV spectra presented in Fig. 1a do not show the presence of such a spectral gap at any impurity level, which contradicts the possible presence of a charge-ordered state.

2. The manuscript lacks convincing experimental evidence that the checkerboard pattern is not arising from quasiparticle bound states, i.e, Yu-Shiba-Rusinov (YSR) states, owing to the interaction between the superconducting condensate and the magnetic Fe impurities. Looking at the presented data, I would go so far that all experimental data presented by the authors, such as filling of the superconducting gap and increasing density of checkerboard pattern etc., could be in principle explained in a picture that solely accounts for an increasing density of YSR states but does not include any speculation on competing orders.

Indeed, the authors present evidence for Fe impurity-induced YSR states that appear as sharp peaks at bias < 10 meV in the dI/dV spectrum presented in Fig. 1f. As the authors write, the length scale of YSR state and checkerboard pattern match. Because the checkerboard pattern already appears at small bias voltages ($V=0, 10, 12$ mV) below the superconducting gap edge (Fig.2, Fig. S3, Fig. S5 etc.), it must, at least at these energies, originate from superconducting quasiparticles. The most plausible origin for this pattern at given measurement conditions (large tunnel conductance) and sub-gap energies are YSR states. In this regard, the competing order picture makes even less sense: If the checkerboard pattern would arise from a competing charge order that dominates the local electronic environment, how can superconducting quasiparticle states still exist in such a competing scenario?

The argument that no checkerboard pattern was found in other work on related materials is obviously insufficient, nor does it come as a surprise that non-magnetic Se impurities do not induce such states. The authors should spatially map out the LDOS peaks arising from the YSR states to exclude these states as a possible origin of the checkerboard pattern. Because the Fermi surface is known, one could even calculate the spatial dependence of the YSR LDOS and compare it with experimental data.

While the Methods section does not provide much detail, it seems that all dI/dV maps were recorded in constant current mode. Because the YSR state appearing at < 10 mV shows a very strong dI/dV

signal, it is conceivable that the observed checkerboard patterns recorded at higher bias voltage set points, e.g., 40 and 50 meV, are nothing but a measurement artefact arising from the so-called set point effect. The fact that the spatial characteristics of this pattern does not change between bias voltage set points below and above the superconducting gap looks rather suspicious to me. This basically means that the spatially resolved LDOS of superconducting and normal quasiparticles is identical. Hence, the authors should present data, such as those taken in a multi-pass measurement scheme, that exclude the set point effect as a possible origin.

3. I don't think that the presented dI/dV maps support the author's claim on nematicity. Even at a lower Fe impurity level (Fig. 3b), one can find areas that locally show a rather unidirectional pattern. The use of quasiparticle scattering seems also questionable in the presence of such a large impurity density ($\sim 20\text{-}30\%$ surface coverage, Fig. 4a).

4. The authors should present data measured on samples with $x > 5\%$. The presence of the checkerboard pattern in the absence of SC could easily exclude the YSR state origin. That is a very easy check.

Reviewer #2 (Remarks to the Author):

Chen et al. study a series of intercalated heavily electron-doped Fe-chalcogenide film samples as a function of Fe defect concentration using scanning tunneling microscopy and spectroscopy. The authors discover a defect-pinned checkerboard charge order (CO) in the sample with the lowest impurity concentration. As the impurity concentration increases, the checkerboard order becomes extended and eventually becomes C_2 -symmetric. This evolution is accompanied by the decrease in superconducting T_c . The authors provide an explanation for the CO using the addition of spin fluctuation wave vectors identified by previously reported neutron scattering studies. The results are very interesting and the data is of good quality. However, I would need the authors to answer the following questions before I can recommend publication of this manuscript.

Regarding the data shown in the paper:

* I am not fully convinced that the 3.2% sample actually shows a unidirectional order (as opposed to also a checkerboard order). Looking at the data presented in Figure 3 and Supplementary Figures 8 and 9 (see for example 10 meV FFT in Figure S9), it seems to me that the $q_{\pm 2\text{Fe}}$ peaks exist along both lattice directions, but a bit stronger along one. This may be due to tip anisotropy. Have the authors observed a domain boundary using the same STM tip with the rotation of unidirectional features by 90 degrees? This would significantly strengthen the authors' claims and rule out tip artifacts.

* In the 0.54% case, some of the Fe dumbbell impurities have the checkerboard pinned to them, and some do not. Can the authors elaborate a bit more why? Are they associated with how much superconductivity is suppressed and/or different in-gap states? The authors could analyze that in a bit more detail to see more clearly if the pinned CO competes with superconductivity or not.

Regarding the explanation of the CO:

* I do not think the use of the term "nematic" is appropriate here, as this in principle implies rotation symmetry breaking without translation symmetry breaking. Here, the translation symmetry is also broken. The authors may consider using terms unidirectional charge ordering or charge stripes.

* It seems to me that the CO stripes observed in FeSe (ref. 20, seems incommensurate) cannot be explained the same way using the addition of spin fluctuation wave vectors (ref. 23, commensurate)? Does that indicate the explanation here is just an individual case and cannot be used to explain the origin of the CO in FeSe?

* It would be nice (but not required) if the authors could explain the evolution of the CO (pinned to extended to "nematic") as a function of Fe defect concentration, and how it is related to the addition of the spin fluctuation wave vectors?

Minor issues:

* In Methods, it says "samples are cleaved...". But is this a thin film grown on a substrate, or a cleaved single crystal?

* Why is the q_{2Fe} charge order peak in pinned checkerboard (0.54%, Fig. 2c) much sharper than the extended checkerboard (2.2%, Fig. 3e)? I am not sure this should be the case if the latter is indeed an extended checkerboard pattern.

Reviewer #3 (Remarks to the Author):

This is an STM study of an electron-doped iron-selenide superconductor. Most other iron-pnictide/iron-selenide superconductors host both electron and hole-type Fermi pockets, and superconductivity is seen to coexist with competing orders. But the present compound has only electron pockets. This paper claims that previous studies have not discovered a coexistence/competition of competing orders with superconductivity in systems with only Fermi surface pockets. This is the first report of such an observation. But this claim is subsequently contrasted by the authors in the introduction that in $KxFe_2Se_2$ compound, which only has an electron-pocket Fermi surface, charge ordering is already observed (Refs. 29,30). In this context, it should also be noted that in electron-doped cuprates, which have only electron-pocket in the antiferromagnetic phase and very similar Fermi surface topology as the present compound, a charge order induced by magnetic order is observed (Nature Physics 15, 335 (2019)). Therefore, the claim of the novelty of this observation is not entirely credible.

Secondly, is the observation of a (short-ranged)-charge/nematic order important and path-breaking in the pnictide field? In my opinion, it is not. The charge order observed here appears to be very short-ranged and mostly pinned near the disorder. Such a parasitic phase is often seen in so many condensed matter systems. It has been unnecessarily overhyped in cuprate and pnictide fields in the last 10-15 years, as it is important to understand superconductivity. But the understanding of superconductivity has not improved a bit. This is because charge fluctuation is always weak and does not contribute to unconventional pairing as the authors also commented on. Therefore, a rediscovery of charge order cannot qualify for publication in Nature Communications.

I have found some obvious problems with this manuscript.

1) The origin of the charge order, as discussed in this paper, is not adequately substantiated. In the one hand, it is postulated to arise from spin-fluctuation and electronic mechanisms. But on the other hand, the charging order is found to be fairly dispersion less. It is often found near the disorder in the measured dI/dV data. But to claim its long-ranged behavior, the authors have taken a QPI spot at the charge order wavevector and inverse-Fourier-transformed the spectrum to real space. This inverse FFT does not incorporate the broadening/spread of the QPI spot and hence will naturally give a long-range pattern. But in reality, the coherence length of the charge order is short ranged. This is also the case in cuprates. The authors should estimate the coherence length of the charge order and plot the coherence length and wavevector as a function of x .

2) Why phonon and/or disorder-induced local strain is ruled out for the origin of charge order? Can one do a Raman experiment to check the softening of any phonon mode?

3) Why is there no x-ray data to observe the charge order? Is the signal too low due to short coherence length? This, at least, should be discussed in the manuscript if the experiment is harder to

conduct.

4) The 'simulation' part is a real bummer. Firstly, the author does not mean theory or calculation when they say 'simulation'. There is no detail of 'simulation' even in the supplementary information. They have considered the 'form-factor' of multiple spin fluctuation wavevectors and summed them with equal weightage. Why equal weightage? Do different wavevectors have the same lifetime and correlation length? Then a 'form-factor' of the charge order is deduced. How?

It's actually not much of an effort to do a mean-field theory or perturbation theory calculation to see if the spin-order is causing the charge order.

Considering these, I think Communications Physics is safe and good for both the authors and Nature journals.

NCOMMS-22-33336-T

Competing orders in heavily electron-doped iron selenide superconductors

by Ziyuan Chen et al.

Reply to the reviewers:

We thank all reviewers for their time and insightful comments on our manuscript. Our point-by-point responses are in blue text below and the original comments are in italic. The corresponding revisions in manuscript are highlighted with yellow background.

Replies to the first reviewer' report:

Reviewer #1

Comment 1: Chen Ziyuan and co-workers report on scanning tunneling microscopy and spectroscopy measurements on the surface of iron (Fe) doped LiFeOHFeSe. Using spectroscopic imaging at different energies they investigate the appearance of localized local density of states (LDOS) patterns in the vicinity of Fe defect sites on the sample surface. Upon increasing Fe defect concentration superconductivity becomes suppressed and the LDOS pattern become more dominant. Comparing their experimental observations with model calculations, the authors argue that Fe defects suppress superconductivity and promote the emergence of competing charge orders whose type depends on the doping level.

Competing orders in cuprates and iron-based superconductors is an active research area and the manuscript under review, therefore, touches on an interesting topic. For reasons that I will describe in the following, the conclusions drawn from the authors appear premature and are not yet sufficiently supported by experimental data in a way that would warrant publication of this work in Nature Communications. The following comments must be addressed before I can recommend publication in Nature Communications.

Reply: We thank the reviewer for the comprehensive summary of the key results of our manuscript and appreciation of the importance of our research. The reviewer puts forward some profound and insightful comments which help us improve our manuscript substantially. Reviewer #1 mainly concerns whether the observed checkerboard pattern is arising from the quasiparticle bound state instead of a competing charge order. We followed the reviewer's suggestion to measure the non-superconducting $(\text{Li}_{0.84}\text{Fe}_{0.16}\text{OH})\text{Fe}_{1-x}\text{Se}$ sample with $x > 5\%$, and successfully observed the presence of a checkerboard order in defect-free FeSe regions, which can definitely rule out the YSR state origin. Below we respond the reviewer's comments point-by-point. We believe our additional measurements and improved analysis/discussion can eliminate the ambiguity in the previous manuscript.

Comment 2: If the increased Fe impurity density were to induce a charge ordered state (nematic or not), a spectral gap should develop in the dI/dV spectrum around Fermi energy. The dI/dV spectra presented in Fig. 1a do not show the presence of such a spectral gap at any impurity level, which contradicts the possible presence of a charge-ordered state.

Reply: We thank the reviewer for raising this point. We found that opening of a spectral gap is actually evidenced in our data, but we didn't make sufficient description on it previously. As shown in revised Fig. 1f, with increased Fe-site defect concentration (x), the superconducting gap is gradually suppressed, as

indicated by weakened coherence peaks and broadening of the gap feature; Meanwhile, the spectral weight within the whole energy range of ± 40 meV is also strongly suppressed with increased x . A broad V-shaped spectrum with low DOS around E_F gradually shows up for high x values, which evidences a spectral gap opening. Such a partial gap is quite common for CDW materials. We added more discussions on the spectral gap induced by checkerboard charge order in the revised manuscript, please see paragraph 2 of page 3 for more details.

Moreover, one of the authors of the present work has participated in the recent angular resolved photoemission spectroscopy (ARPES) measurements on $(\text{Li}_{0.84}\text{Fe}_{0.16}\text{OH})\text{Fe}_{1-x}\text{Se}$ samples with varying x values, which also observed spectral weight suppression and gap-like feature around E_F with higher x values (private communications and the ARPES paper is under review). The gap size obtained by ARPES is about 20 ~ 40 meV for samples with varying T_c values, consistent with the energy scale of V-shaped DOS suppression here.

Comment 3: *The manuscript lacks convincing experimental evidence that the checkerboard pattern is not arising from quasiparticle bound states, i.e., Yu-Shiba-Rusinov (YSR) states, owing to the interaction between the superconducting condensate and the magnetic Fe impurities. Looking at the presented data, I would go so far that all experimental data presented by the authors, such as filling of the superconducting gap and increasing density of checkerboard pattern etc., could be in principle explained in a picture that solely accounts for an increasing density of YSR states but does not include any speculation on competing orders.*

Indeed, the authors present evidence for Fe impurity-induced YSR states that appear as sharp peaks at bias < 10 meV in the dI/dV spectrum presented in Fig. 1f. As the authors write, the length scale of YSR state and checkerboard pattern match. Because the checkerboard pattern already appears at small bias voltages ($V=0, 10, 12$ mV) below the superconducting gap edge (Fig.2, Fig. S3, Fig. S5 etc.), it must, at least at these energies, originate from superconducting quasiparticles. The most plausible origin for this pattern at given measurement conditions (large tunnel conductance) and sub-gap energies are YSR states. In this regard, the competing order picture makes even less sense: If the checkerboard pattern would arise from a competing charge order that dominates the local electronic environment, how can superconducting quasiparticle states still exist in such a competing scenario?

The argument that no checkerboard pattern was found in other work on related materials is obviously insufficient, nor does it come as a surprise that non-magnetic Se impurities do not induce such states. The authors should spatially map out the LDOS peaks arising from the YSR states to exclude these states as a possible origin of the checkerboard pattern. Because the Fermi surface is known, one could even calculate the spatial dependence of the YSR LDOS and compare it with experimental data.

While the Methods section does not provide much detail, it seems that all dI/dV maps were recorded in constant current mode. Because the YSR state appearing at < 10 mV shows a very strong dI/dV signal, it is conceivable that the observed checkerboard patterns recorded at higher bias voltage set points, e.g., 40 and 50 meV, are nothing but a measurement artefact arising from the so-called set point effect. The fact that the spatial characteristics of this pattern does not change between bias voltage set points below and above the superconducting gap looks rather suspicious to me. This basically means that the spatially resolved LDOS of superconducting and normal quasiparticles is identical. Hence, the authors should present data, such as those taken in a multi-pass measurement scheme, that exclude the set point effect as a possible origin.

Reply: We agree with reviewer that alternative interpretations of the data shall be carefully examined. Following reviewer's suggestions, we have carried out additional measurements to verify the origin of checkerboard pattern and the effect of YSR bound state. We will show below that the alternative interpretation based on YSR state can be ruled out.

Firstly, we measured the non-superconducting $(\text{Li}_{0.84}\text{Fe}_{0.16}\text{OH})\text{Fe}_{1-x}\text{Se}$ with $x > 5\%$ to check the presence of the checkerboard pattern, which is a direct way to exclude the YSR state origin. We carried out STM study on the non-superconducting sample #4 (with its properties shown in Supplementary Fig. S1). The amount of Fe-site defects increases a lot comparing to the superconducting samples, and a broad V-shaped spectrum is observed around E_F (the magenta curve in revised Fig. 1f). We have changed several measurement conditions to obtain dI/dV maps, and they all show similar static checkerboard pattern in defect-free FeSe regions within the energy range of about $-40 \sim +70$ meV. Moreover, for all $(\text{Li}_{0.84}\text{Fe}_{0.16}\text{OH})\text{Fe}_{1-x}\text{Se}$ samples, we find the checkerboard pattern is most pronounced at $E = 30\sim 50$ meV, which is not at the energies of YSR state peaks or in-side of superconducting gap. These results can definitely rule out the YSR state origin, and also exclude the set point effect.

Besides, we studied the spatial distribution of YSR states induced by the most commonly observed type-I Fe-site defects by using high-resolution STM/STS. As show in Fig. S5 of SM, two pairs of YSR states located at ± 3.1 meV and ± 5.0 meV are observed. They display long-range LDOS oscillations with a wave vector of $2k_F$ (period ≈ 1.9 nm), and persists up to ~ 10 nm away from the defect. The oscillation phase shift is different for different YSR states, which is clearly distinguished from the static checkerboard pattern with a period of $2a_{Fe} \approx 0.54$ nm. Moreover, at ultralow temperature of 20 mK, these YSR states induced LDOS modulation is only significant at the energies of YSR peaks. At 4.2 K, the YSR peaks are broader, but their spatial distribution does not change. Therefore, the YSR state behaves differently from the observed checkerboard charge order in both spatial distribution and energy scale, thus excluding it as the possible origin of the checkerboard order.

As for the coexistence of checkerboard pattern and superconducting quasiparticle states, we notice that it has been widely observed in cuprates, e.g., pronounced checkerboard pattern was originally observed in vortex core of optimally doped BSCCO (ref. 8), and then for a wide doping range (Science 303, 1995 (2004); Nature 430, 1001 (2004); PRL, 94, 197005 (2005); Nat. phys. 4, 696 (2008)). Recently, local AFM order is observed near the Fe defect in BSCCO (PNAS 118, e2115317118(2021)). In our case, the checkerboard order arises when the superconductivity is suppressed by disorder nearby, although it may not be completely suppressed. Such a competition does not rule out the coexistence of both orders.

We added the new datasets on non-superconducting $(\text{Li}_{0.84}\text{Fe}_{0.16}\text{OH})\text{Fe}_{1-x}\text{Se}$ in the revised manuscript, please see Fig. 1, Fig. 4, Supplementary Fig. S12, Fig. S13 and related descriptions in the revised text for more details. Moreover, the spatial distribution of YSR states induced by the most commonly observed type-I Fe-site defects were also added in Fig. S5 and discussed in the revised text.

Comment 4: *I don't think that the presented dI/dV maps support the author's claim on nematicity. Even at a lower Fe impurity level (Fig. 3b), once can find areas that locally show a rather unidirectional pattern. The use of quasiparticle scattering seems also questionable in the presence of such a large impurity density ($\sim 20\text{-}30\%$ surface coverage, Fig.4a).*

Reply: Based on our new measurements on non-superconducting samples, we would agree with reviewer on this point that the stripe-like pattern is not a well-established order. We find that a checkerboard order is also present in the defect-free FeSe areas of non-superconducting $(\text{Li}_{0.84}\text{Fe}_{0.16}\text{OH})\text{Fe}_{1-x}\text{Se}$ films (revised Fig. 4), while unidirectional charge stripes only appear in the immediate vicinity of some Fe-site defect clusters, as shown in Supplementary Fig. S14. Therefore, the C_4 -symmetric checkerboard order is the dominate competing order and exists over the whole phase space of heavily electron-doped FeSe materials; while the appearance of unidirectional charge stripes strongly depend on the local environment. We note

that unidirectional charge stripes have been observed in several-layer FeSe films grown on SrTiO₃ or BaTiO₃ substrates, but are more pronounced in the latter (Nat. Phys. 13, 957 (2017); Nat. Commun. 12, 41467 (2021); Phys. Rev. B 106, 024517 (2022)). The increased tensile strain in FeSe/BTO than FeSe/STO leads to stronger electronic anisotropy and correlation. In our case, unidirectional charge stripes can only be occasionally observed near the immediate vicinity of Fe-site defect clusters; we suspect that the gathered Fe-site defects in a small sample region may distort the lattice locally and induce a local strain, which is responsible for the appearance of local charge stripes. Such speculation needs further investigation. Accordingly, we revise related discussions in the revised manuscript, and use unidirectional charge stripes to replace the statement of nematic charge order. Please see paragraph 2 in page 7 of the main text and part 6 of Supplementary materials for more details.

We also agree that the use of quasiparticle scattering is not appropriate when the impurity density is high, we change the description from quasiparticle scattering to local density of state distribution to avoid misunderstanding.

Comment 5: *The authors should present data measured on samples with $x > 5\%$. The presence of the checkerboard pattern in the absence of SC could easily exclude the YSR state origin. That is a very easy check.*

Reply: We thank the reviewer for this excellent suggestion. As we discussed in the reply to comment 3, we did carry out STM measurements on non-superconducting (Li_{0.84}Fe_{0.16}OH)Fe_{1-x}Se sample with $x > 5\%$ and indeed observe a similar checkerboard charge order in defect-free FeSe regions, which can definitely rule out the YSR state origin and greatly enhance the credibility of a competing checkerboard charge order. Please see the above reply to comment 3 and the revised manuscript for more details.

Replies to the second reviewer' report:

Reviewer #2

Comment 1: *Chen et al. study a series of intercalated heavily electron-doped Fe-chalcogenide film samples as a function of Fe defect concentration using scanning tunneling microscopy and spectroscopy. The authors discover a defect-pinned checkerboard charge order (CO) in the sample with the lowest impurity concentration. As the impurity concentration increases, the checkerboard order becomes extended and eventually becomes C2-symmetric. This evolution is accompanied by the decrease in superconducting T_c. The authors provide an explanation for the CO using the addition of spin fluctuation wave vectors identified by previously reported neutron scattering studies. The results are very interesting and the data is of good quality. However, I would need the authors to answer the following questions before I can recommend publication of this manuscript.*

Reply: We thank the reviewer for the comprehensive summary of the key results of our manuscript and giving high regard of our work. Below we respond to the questions the reviewer concerned point-by-point.

Comment 2: **I am not fully convinced that the 3.2% sample actually shows a unidirectional order (as opposed to also a checkerboard order). Looking at the data presented in Figure 3 and Supplementary Figures 8 and 9 (see for example 10 meV FFT in Figure S9), it seems to me that the q_{2Fe} peaks exists along both lattice directions, but a bit stronger along one. This may be due to tip anisotropy. Have the authors observed a domain boundary using the same STM tip with the rotation of unidirectional features by 90 degrees? This would significantly strengthen the authors' claims and rule out tip artifacts.*

Reply: We thank the reviewer for checking our data very carefully and pointing out this issue. To see if there is a transition from checkerboard to stripe charge orders with increasing x values, we carried out STM study on the non-superconducting sample #4 with $x \sim 6.35\%$ (with its properties shown in Fig. 1 and Fig. S1). The dI/dV maps show the presence of a similar checkerboard charge order in defect-free FeSe regions, while the unidirectional charge stripes are not obvious (Fig. 4 and Supplementary Fig. S12, S13). We also reanalysed the dI/dV maps for different x values carefully, and find that the unidirectional charge stripes do exist but usually localized in the immediate vicinity of some Fe-site defect clusters, while the other regions remain the C_4 -symmetric checkerboard patterns, as shown in Supplementary S14. In Fig. S14b, the directions of the charge stripes in different sample regions rotate by 90 degrees, excluding the tip effect. Therefore, we conclude that the C_4 -symmetric checkerboard order is the dominated competing order over the whole phase space of heavily electron-doped FeSe, while the appearance of unidirectional charge stripes strongly depend on the local environment. Accordingly, we revised related discussions in the manuscript, and use unidirectional charge stripes to replace the statement of nematic charge order. Please see paragraph 2 in page 7 of the main text and part 6 of Supplementary materials for more details.

Comment 3: ** In the 0.54% case, some of the Fe dumbbell impurities have the checkerboard pinned to them, and some do not. Can the authors elaborate a bit more why? Are they associated with how much superconductivity is suppressed and/or different in-gap states? The authors could analyze that in a bit more detail to see more clearly if the pinned CO competes with superconductivity or not.*

Reply: We thank the reviewer for pointing out this issue. According to earlier studies [PRB **94**, 134502 (2016); PRB **97**, 024502 (2018); PRB 100,155127 (2019); Communications Physics 3,75 (2020)] and our experience, the dumbbell-shaped defects can be classified into at least two types, and they all suppress superconductivity. In our study, the Fe-site defects can be roughly divided into two types according to the impurity potential, as shown in Fig. 2 and Supplementary Fig. S2. The type-I Fe-site defects, probably Fe vacancies (V_{Fe}), are the most commonly observed and exhibit a strong impurity scattering potential; they strongly suppress the superconductivity and induce sharp in-gap YSR states (blue curve in Fig. S2c), and obvious QPI patterns appears around them (Fig. S2b). The type-II Fe-site defects possess a much weaker impurity scattering potential and mainly affect the superconducting state at gap edges (red curve in Fig. S2c), the surrounding QPI signals are very weak compared with type-I defects. We conclude that both types of Fe-site defects suppress superconductivity and pin the checkerboard pattern. For dI/dV maps shown in Fig. 2 and Supplementary Fig. S4, it is obvious that the checkerboard pattern is pinned at the type-II Fe-site defects; while it is not so certain for the type-I defects due to the surrounding strong LDOS oscillations which may overwhelm the weak checkerboard pattern, the iFFT images after filtering out QPI signals indeed show pinned checkerboard patterns at both the type-I and type-II defects, but not at Se vacancies or defect-free FeSe areas where the superconductivity is not affected. In the revised manuscript, we added related discussions, please see Fig. 2 of the main text, Fig. S2 and S4 of Supplementary materials and related descriptions for more details.

Comment 4: *I do not think the use of the term “nematic” is appropriate here, as this in principle implies rotation symmetry breaking without translation symmetry breaking. Here, the translation symmetry is also broken. The authors may consider using terms unidirectional charge ordering or charge stripes.*

Reply: We thank the reviewer for pointing out this issue. We agree with reviewer that the term ‘nematic’ is not that appropriate here, so we use the term unidirectional charge stripes instead in the revised version.

Comment 5: *It seems to me that the CO stripes observed in FeSe (ref. 20, seems incommensurate) cannot be explained the same way using the addition of spin fluctuation wave vectors (ref. 23, commensurate)?*

Does that indicate the explanation here is just an individual case and cannot be used to explain the origin of the CO in FeSe?

Reply: The electronic structures of bulk FeSe and several-layer FeSe/STO or FeSe/BTO are very different from the case here. Moreover, the magnetic excitations of bulk FeSe (with both electron and hole Fermi surfaces) and heavily electron doped FeSe (with only electron Fermi surfaces) are remarkably different (Refs. 23, 31 and 32). In addition, it is known that strain can alter these properties even further, so one may not use the spin fluctuations measured in bulk FeSe (ref. 23) to explain the CO stripes measured in thin films (ref. 20). In any case, the CO stripes in ref. 20 may not be explained in the same way used here without the accurate spin fluctuation information of these systems.

Comment 6: *It would be nice (but not required) if the authors could explain the evolution of the CO (pinned to extended to “nematic”) as a function of Fe defect concentration, and how it is related to the addition of the spin fluctuation wave vectors?*

Reply: The evolution from pinned to extended checkerboard charge pattern arises from the enhanced suppression of superconductivity by increased x . As explained in the above reply to your comment 2, and the reply to the comment 4 of the 1st reviewer, we now consider the evolution between checkerboard and stripe order is likely induced by local strains induced by the accumulation of Fe-site defects. We have added related discussion in paragraphs 2 and 3 in page 7 of the revised manuscript.

For the relations with spin fluctuations, we assign the checkerboard and stripe orders as the superposition of different spin fluctuations with equal or unequal weightage, respectively. As seen in Fig. 5a, the low energy magnetic excitations of $\vec{Q}_1, \vec{Q}_2, \vec{Q}'_1, \vec{Q}'_2, \vec{Q}_3, \vec{Q}_4, \vec{Q}'_3, \vec{Q}'_4$ found by INS measurements on optimally doped $(\text{Li}_{0.84}\text{Fe}_{0.16}\text{OD})\text{Fe}_{1-x}\text{Se}$ (ref. 31) show similar spectral weights. The horizontal $\vec{Q}_3, \vec{Q}_4, \vec{Q}'_3, \vec{Q}'_4$ and vertical $\vec{Q}_1, \vec{Q}_2, \vec{Q}'_1, \vec{Q}'_2$ patterns correspond to the scatterings between the Fermi surface sectors of d_{xy} orbitals with that of d_{xz} or d_{yz} orbitals, respectively. In heavily electron-doped FeSe systems, the lattice is C_4 -symmetric and the d_{xz} or d_{yz} orbitals are degenerate, thus the nesting between d_{xy} orbitals and d_{xz} or d_{yz} orbitals are degenerate as well, leading to the same spectral weights of different spin fluctuations. The addition of multiple spin fluctuation wavevectors with equal weightage leads to the observation of checkerboard order. If the degeneracy of d_{xz} and d_{yz} orbitals is destroyed, such as by local strain, unequal weightage should be considered and corresponds to the appearance of charge stripes. This discussion has been added into the revised manuscript, see paragraph 1 of page 10 for more details.

Comment 7: *In Methods, it says “samples are cleaved...”. But is this a thin film grown on a substrate, or a cleaved single crystal?*

Reply: Our samples are high-quality single-crystalline $(\text{Li}_{0.84}\text{Fe}_{0.16}\text{OH})\text{Fe}_{1-x}\text{Se}$ films grown on a LaAlO_3 substrate by a hydrothermal epitaxial method. They have a typical thickness of ~ 100 nm, and can be cleaved for several times. Related discussion in Methods section was modified.

Comment 8: *Why is the $q_{2\text{Fe}}$ charge order peak in pinned checkerboard (0.54%, Fig. 2c) much sharper than the extended checkerboard (2.2%, Fig. 3e)? I am not sure this should be the case if the latter is indeed an extended checkerboard pattern.*

Reply: We thank the reviewer for pointing out this issue. We can see that the intensity of $q_{2\text{Fe}}$ peak in Fig. 2c is much weaker due to the low concentration of the pinned checkerboard domains, while it is much stronger in Fig. 3e due to the enhanced concentration of the checkerboard pattern. As seen in Fig. 3 and Supplementary Fig. S7, the checkerboard pattern indeed extends to the defect-free FeSe regions, but it is

not a long-range correlated order. Due to the existence of amounts of defects, the checkerboard pattern is short-ranged and shows strong disorder, which leads to the broadening of the q_{2Fe} peak in Fig. 3e.

Replies to the third reviewer' report:

Reviewer #3

Comment 1: *This is an STM study of an electron-doped iron-selenide superconductor. Most other iron-pnictide/iron-selenide superconductors host both electron and hole-type Fermi pockets, and superconductivity is seen to coexist with competing orders. But the present compound has only electron pockets. This paper claims that previous studies have not discovered a coexistence/competition of competing orders with superconductivity in systems with only Fermi surface pockets. This is the first report of such an observation. But this claim is subsequently contrasted by the authors in the introduction that in $K_xFe_2Se_2$ compound, which only has an electron-pocket Fermi surface, charge ordering is already observed (Refs. 29,30). In this context, it should also be noted that in electron-doped cuprates, which have only electron-pocket in the antiferromagnetic phase and very similar Fermi surface topology as the present compound, a charge order induced by magnetic order is observed (Nature Physics 15, 335 (2019)). Therefore, the claim of the novelty of this observation is not entirely credible.*

Reply: We understand reviewer's concern. As reviewer mentioned, in the introduction section we have stated that 2×2 charge order had been observed in heavily electron-doped $K_{1-x}Fe_{2-y}Se_2$, but its relationship to superconductivity and its origin had not been studied in detail. In our study, we do not claim to be the first report of competing order in heavily electron-doped iron-selenide, but focus on the discovery of intertwined orders in a large phase space from optimal doping to non-superconducting regions (tuned by Fe-site defects), as well as their origin and relationship with superconductivity. Particularly, the wave vector of observed charge patterns is quantitatively consistent with the multi-Q spin fluctuation obtained by neutron scattering. We therefore attribute the origin of charge ordering to magnetic fluctuations. We think this study greatly facilitate the understanding of pairing mechanism of heavily electron-doped FeSe, and reveals their similarities/differences with other iron-based superconductors. Therefore, our study would have significant novelty.

Comment 2: *Secondly, is the observation of a (short-ranged)-charge/nematic order important and path-breaking in the pnictide field? In my opinion, it is not. The charge order observed here appears to be very short-ranged and mostly pinned near the disorder. Such a parasitic phase is often seen in so many condensed matter systems. It has been unnecessarily overhyped in cuprate and pnictide fields in the last 10-15 years, as it is important to understand superconductivity. But the understanding of superconductivity has not improved a bit. This is because charge fluctuation is always weak and does not contribute to unconventional pairing as the authors also commented on. Therefore, a rediscovery of charge order cannot qualify for publication in Nature Communications.*

Reply: We understand that the relationship between charge orders and superconductivity in cuprates may be still unclear and under debate, however, it may be premature to deny the vast studies on them yet. As the competing orders occupied a large portion of phase diagram of both cuprate and iron-based superconductors, they cannot be ignored without careful checking their effects on superconductivity and the relations to the pairing glue.

Although we observed the charge order, our analysis suggests that it is probably driven by spin fluctuations. Such a short-range checkerboard charge order persists from optimal superconductivity to non-superconducting regimes. Considering a similar checkerboard charge order observed in heavily electron-

doped $K_{1-x}Fe_{2-y}Se_2$, such a charge order may be a universal characteristic of heavily electron-doped FeSe systems. Our data highlight the two faces of spin fluctuations, which are important for both superconducting pairing and static order, consistent with the observations made in cuprates and other iron-based superconductors.

Comment 3: *The origin of the charge order, as discussed in this paper, is not adequately substantiated. In the one hand, it is postulated to arise from spin-fluctuation and electronic mechanisms. But on the other hand, the charging order is found to be fairly dispersionless. It is often found near the disorder in the measured dI/dV data. But to claim its long-ranged behavior, the authors have taken a QPI spot at the charge order wavevector and inverse-Fourier-transformed the spectrum to real space. This inverse FFT does not incorporate the broadening/spread of the QPI spot and hence will naturally give a long-range pattern. But in reality, the coherence length of the charge order is short ranged. This is also the case in cuprates. The authors should estimate the coherence length of the charge order and plot the coherence length and wavevector as a function of x .*

Reply: There could be some misunderstanding due to our insufficient description in the previous manuscript. The using of the term ‘extended checkerboard pattern’ does not refer to a long-ranged order but just distinguish it from the pinned checkerboard pattern shown in Fig. 2. Our main purpose of using iFFT is to show where the checkerboard patterns exist. We agree with the reviewer that the extended checkerboard pattern is short-ranged, as can be seen clearly in the dI/dV maps and iFFT images shown in part 5 of SM. Our iFFT process actually considered the broadening/spread of the q_{2Fe} spots and do show a short-ranged charge order.

By carrying out iFFT with intensities near q_{2Fe} and then getting the autocorrelation maps for the regions shown in the text following the methods described in Phys. Rev. B 89, 235115 (2014), we find that the in-plane correlation lengths of the observed charge orders are all close to $4a_{Fe} \sim 5a_{Fe}$ for varying x .

A static order is always dispersionless as a function of energy by definition. We assume the reviewer meant the doping independency by “dispersionless”. In our study we find that the Fe-site defect concentration can regulate the superconductivity effectively, but has little influence on carrier doping and the whole band structures, as indicated in Fig. 1e. Thus it is reasonable to conclude that the scattering between different parts of the Fermi surface, the induced spin fluctuations and charge patterns do not depend much on doping x .

In the revised manuscript, we estimated the in-plane correlation lengths of the short-ranged charge orders, and added related discussions in the main text and SM files, please see paragraph 3 of page 7 and Supplementary Fig. S15 for more details.

Comment 4: *Why phonon and/or disorder-induced local strain is ruled out for the origin of charge order? Can one do a Raman experiment to check the softening of any phonon mode?*

Reply: we thank the reviewer for pointing out other possibilities for the origin of charge order.

As for the possible phonon origin.

For conventional charge orders arising from electron-phonon coupling, the charge modulations can generally be seen in the STM topographic images (Sur. Sci. Rep. 76, 100523 (2021), JPCM 14, 8393 (2002)). We did not detect $2a_{Fe}$ modulations in STM topographic images at any x . Thus we think the phonon origin is unlikely.

As for whether the disorder-induced local strain causes the checkerboard charge order.

Since the checkerboard pattern already exists locally on the isolated Fe-site defect in optimally doped samples, and extends to the large defect-free FeSe regions in the samples with $x=1.8\%$ and 2.2% , as shown in Fig. 3 and Fig. S9, where there are not so many defects and the induced local strain should not be too strong. Moreover, with the increased x values, disorder-induced local strain will increase in principle, but the observed checkerboard pattern does not change. Therefore, we think it can be ruled out.

The above discussions have been added to the revised manuscript, please see paragraph 3 in page 11 for more details.

To check the phonon softening at the finite q requires IXS or INS, as Raman is a $q=0$ measurement. It will be very challenging for such thin film samples at the moment, which is unfortunately beyond the scope of the current work. Raman signal from thin samples will be very weak as well.

Comment 5: *Why is there no x-ray data to observe the charge order? Is the signal too low due to short coherence length? This, at least, should be discussed in the manuscript if the experiment is harder to conduct.*

Reply: Yes, we think it will be not so easy to measure the charge order in $(\text{Li}_{0.84}\text{Fe}_{0.16}\text{OH})\text{Fe}_{1-x}\text{Se}$ by x-ray experiments. The main reason is the short-ranged and strong disordered behavior of the charge order, as well as the thin film samples, which may lead to a very low signal. Besides, the temperature at which the charge order occurs is unclear now, and it may be necessary to reach a low temperature to see the charge order signals, which is also a limitation. We follow the reviewer's suggestion and propose the suggestion of x-ray measurements on the charge order in the revised text, expecting the professional x-ray teams to try in the future. Please see paragraph 3 in page 11 for more details.

Comment 6: *The 'simulation' part is a real bummer. Firstly, the author does not mean theory or calculation when they say 'simulation'. There is no detail of 'simulation' even in the supplementary information. They have considered the 'form-factor' of multiple spin fluctuation wavevectors and summed them with equal weightage. Why equal weightage? Do different wavevectors have the same lifetime and correlation length? Then a 'form-factor' of the charge order is deduced. How?*

Reply: Our simulation is a simple coherent superposition of spin density waves, but it can qualitatively reproduce the observed charge orders well. Therefore we think it caught, at least partially, the physics behind charge order. We will explain this more explicitly in the revised manuscript.

For the using of equal weighted Q vectors: Firstly, the low energy magnetic excitations at $(\pi \pm \delta\pi, \pi)$ and $(\pi, \pi \pm \delta\pi)$ ($\delta \sim 0.38$) found by INS measurements on optimally doped $(\text{Li}_{0.84}\text{Fe}_{0.16}\text{OD})\text{Fe}_{1-x}\text{Se}$ (ref. 31) show similar spectral weights. The horizontal $\vec{Q}_3, \vec{Q}_4, \vec{Q}'_3, \vec{Q}'_4$ and vertical $\vec{Q}_1, \vec{Q}_2, \vec{Q}'_1, \vec{Q}'_2$ patterns correspond to the scatterings between the Fermi surface sectors of d_{xy} orbitals with that of d_{xz} or d_{yz} orbitals, respectively. In heavily electron-doped FeSe systems, the lattice is C_4 -symmetric and the d_{xz} or d_{yz} orbitals are degenerate, thus the nesting between d_{xy} orbitals and d_{xz} or d_{yz} orbitals are degenerate as well, leading to the same spectral weights of different spin fluctuations. That is why we considered the addition of multiple spin fluctuation wavevectors with equal weightage. If the degeneracy of d_{xz} and d_{yz} orbitals is destroyed, such as by the local strain, we choose unequal weightage. These discussion has been added into the revised manuscript, see paragraph 1 in page 9 for more details.

Comment 7: *It's actually not much of an effort to do a mean-field theory or perturbation theory*

calculation to see if the spin-order is causing the charge order.

Reply: It is known that spin order can drive charge order, for examples, in bulk Cr (RMP 60, 209 (1988)), cuprates (RMP 75, 1201 (2003); PRB 66, 094501 (2002)), MnP (ref. 63), and predicted for iron-based superconductors (PRB 82, 144522 (2010)). In all these cases, the period of the induced charge modulation is half of that of the spin modulation. In a mean field theory, depending on what one adds in, such a case can be certainly reproduced once again. However, we do not think it will add much weight to our handwaving argument, since mean field theory cannot accurately describe such a strongly correlated material. So we decide to leave that for interested theorists.

We thank the reviewer for the thoughtful comments. We hope that we have clarify some of his/her doubts. With the revisions, we hope he/she will find the manuscript significantly improved.

REVIEWERS' COMMENTS

Reviewer #1 (Remarks to the Author):

I appreciate the additional experimental efforts undertaken by Chen Ziyuan and co-workers to address my comments. I do share Reviewer 3's opinion that charge order is a rather overhyped topic that—at least so far—has contributed little to understand superconductivity in various compounds and may actually be an unrelated phenomenon. However, it still deserves scrutiny as the authors write and, as shown in this manuscript, may be used as a tool to detect spin fluctuations in the underlying parent phase. In my opinion, this is the key insight of this study, which the authors may want to highlight, instead of talking about competing orders (is kind of pointless after 15 years of seeing similar results in other materials as Reviewer 3 writes). For example, the title does not accurately represent the key results despite sounding flashy.

Overall, the revised manuscript is of much improved quality and should be of interest to research community on iron based superconductors. I can, therefore, recommend publication in Nature Communications.

Reviewer #2 (Remarks to the Author):

The authors have addressed the majority of my main concerns. In particular, the claims of "nematicity" have been toned down and the paper is much more solid now. I am happy to give my recommendation to publish.

Reviewer #3 (Remarks to the Author):

I appreciate the author's effort to address some of the concerns I raised in the previous report. My main concern about the novelty of this work and its importance to the origin of superconductivity remain valid, and the authors could not overturn them. Previously, a charge order was observed in superconductors hosting only electron pockets, which the authors now acknowledge for the selenides (e.g., by revising the abstract as "charge/spin order has not been identified" to "charge/spin order has rarely been identified"), but somehow chose not to acknowledge for the cuprates.

The authors responded as "we think this study greatly facilitate the understanding of pairing mechanism of heavily electron-doped FeSe". This is just a slogan, void of any scientific justification.

I am generally satisfied with the responses to the technical comments. In conclusion, I think the paper is publishable in another journal (such as Communication Physics), but it does meet the criterion of novelty or importance to be published in Nature Communications.

NCOMMS-22-33336B

Competing orders in heavily electron-doped iron selenide superconductors

by Ziyuan Chen et al.

Reply to the reviewers:

We thank all reviewers for their time and insightful comments on our manuscript. Our point-by-point responses are in blue text below and the original comments are in italic. The corresponding revisions in manuscript are highlighted with yellow background.

Replies to the first reviewer' report:

Reviewer #1

Comment 1: I appreciate the additional experimental efforts undertaken by Chen Ziyuan and co-workers to address my comments. I do share Reviewer 3's opinion that charge order is a rather overhyped topic that—at least so far—has contributed little to understand superconductivity in various compounds and may actually be an unrelated phenomenon. However, it still deserves scrutiny as the authors write and, as shown in this manuscript, may be used as a tool to detect spin fluctuations in the underlying parent phase. In my opinion, this is the key insight of this study, which the authors may want to highlight, instead of talking about competing orders (is kind of pointless after 15 years of seeing similar results in other materials as Reviewer 3 writes). For example, the title does not accurately represent the key results despite sounding flashy.

Reply: We thank the reviewer for the positive evaluation of our revised manuscript. We agree with the reviewer's opinion that the charge order itself may be not so important but it can be used as a tool to detect the underlying spin fluctuations, which is the key insight of our study. We followed this advice and revised the title, abstract, introduction and discussion parts of our manuscript to weaken the statement of competing charge order and to emphasize the use of charge order to detect spin fluctuations. The title of our manuscript was changed to "Multiple-Q spin fluctuations driven charge order in heavily electron-doped iron selenide superconductors", and please see the abstract, introduction and discussion parts of the revised manuscript for more revisions.

Comment 2: Overall, the revised manuscript is of much improved quality and should be of interest to research community on iron based superconductors. I can, therefore, recommend publication in Nature Communications.

Reply: We thank the reviewer's recommendation for the publication of our manuscript in Nature communications, and we also appreciate the expertise and time the reviewer provided during last several months.

Replies to the second reviewer' report:

Reviewer #2

Comment 1: The authors have addressed the majority of my main concerns. In particular, the claims of "nematicity" have been toned down and the paper is much more solid now. I am happy to give my recommendation to publish.

Reply: We thank the reviewer's recommendation for the publication of our manuscript in Nature communications, and we also appreciate the expertise and time the reviewer provided during last several months.

Replies to the third reviewer' report:

Reviewer #3

Comment 1: I appreciate the author's effort to address some of the concerns I raised in the previous report. My main concern about the novelty of this work and its importance to the origin of superconductivity remain valid, and the authors could not overturn them. Previously, a charge order was observed in superconductors hosting only electron pockets, which the authors now acknowledge for the selenides (e.g., by revising the abstract as "charge/spin order has not been identified" to "charge/spin order has rarely been identified"), but somehow chose not to acknowledge for the cuprates.

Reply: We thank the reviewer's positive evaluation of our responses in last revised manuscript, and we understand his/her main concern about the novelty of this work by focusing to discuss the competing charge order. This concern has also been mentioned by the first reviewer and he/she suggested us to change the key point of our manuscript to that the observed charge order can be used as a tool to detect spin fluctuations in the parent phase. We appreciate these suggestions and accordingly revised the title, abstract, introduction and discussion parts of our manuscript to weaken the statement of competing charge order and to emphasize the use of charge order to detect spin fluctuations. The title of our manuscript was changed to "Multiple-Q spin fluctuations driven charge order in heavily electron-doped iron selenide superconductors", and please see the abstract, introduction and discussion parts of the revised manuscript for more revisions. We hope our revisions can eliminate this concern.

Besides, the reviewer reminded us to acknowledge the related works about cuprates with only electron pockets. We are sorry we missed that, but it was not our intention. We have cited the related paper (ref. 12 in the main text) in the revised manuscript.

Comment 2: The authors responded as "we think this study greatly facilitate the understanding of pairing mechanism of heavily electron-doped FeSe". This is just a slogan, void of any scientific justification.

Reply: We thank the reviewer's comment. In the revised manuscript, we have revised or removed these statements to make them more accurate, please see the abstract and the last paragraph of discussion part for more details.

Comment 3: I am generally satisfied with the responses to the technical comments. In conclusion, I think the paper is publishable in another journal (such as Communication Physics), but it does not meet the criterion of novelty or importance to be published in Nature Communications.

Reply: We thank the reviewer's recommendation for the publication of our manuscript in Nature Communications, and we also appreciate the expertise and time the reviewer provided during last several months.